# Neural Neural Textures Make Sim2Real Consistent

**Ryan Burgert**   **Jinghuan Shang**   **Xiang Li**   **Michael S. Ryoo**
Department of Computer Science
Stony Brook University
Stony Brook, NY 11794
`{rburgert,jishang,xiangli8,mryoo}@cs.stonybrook.edu`

**Abstract:** Unpaired image translation algorithms can be used for sim2real tasks, but many fail to generate temporally consistent results. We present a new approach that combines differentiable rendering with image translation to achieve temporal consistency over indefinite timescales, using surface consistency losses and *neural neural textures*. We call this algorithm TRITON (Texture Recovering Image Translation Network): an unsupervised, end-to-end, stateless sim2real algorithm that leverages the underlying 3D geometry of input scenes by generating realistic-looking learnable neural textures. By settling on a particular texture for the objects in a scene, we ensure consistency between frames statelessly. TRITON is not limited to camera movements — it can handle the movement and deformation of objects as well, making it useful for downstream tasks such as robotic manipulation. We demonstrate the superiority of our approach both qualitatively and quantitatively, using robotic experiments and comparisons to ground truth photographs. We show that TRITON generates more useful images than other algorithms do. Please see our project website: tritonpaper.github.io

**Keywords:** sim2real, image translation, differentiable rendering

## 1  Introduction

Current sim2real image translation algorithms used for robotics [1, 2] often have difficulty generating consistent results across large time-frames particularly when the objects in an environment are allowed to move. This makes training good robotic policies using such sim2real challenging. In this paper, we discuss an algorithm called TRITON (Texture Recovering Image Translation Network) that combines neural rendering, image translation, and two special surface consistency losses to create surface-consistent translations over frames. We use the term surface-consistent to refer to the desirable quality of preserving the visual appearance of object surfaces as they move, or are viewed from another angles.

TRITON is applicable when we have a simulator capable of generating a realistic distribution of geometric data, but when we do not have any information about the surfaces of those objects (which we need to render realistic images). For example, in one of our experiments we use a robotic arm model provided by the manufacturer that contains no material information, and then learn to render the materials of the arm using TRITON. As opposed to requiring an expert/artist to create 3D models with meaningful textures, TRITON can generate these from scratch using a set of photographs capturing the distribution of states in a simulation, without requiring any matches between domains. That is, given a set of unpaired, unannotated real images and geometry images, TRITON learns the underlying texture of objects and surfaces appearing in the scene without any direct supervision.

We introduce the concept of *neural neural texture*, which is an implicit texture representation having the form of a neural network function generating RGB values given surface coordinates. Unlike previous raster-based 'neural textures' to create realistic images [3, 4], we model surface textures as a function instead of discretized pixels. It is a continuous implicit pixel-less parametric texture represented by a neural network using Fourier features [5] that takes in 2D UV coordinates and outputs colors. This is similar to how neural radiance fields (NeRF) represents a 3D scene in the form of a neural function [6]. The difference is that we focus on learning textures from unpaired training images of very different scene configurations, for viewpoint as well as object motion synthesis.

6th Conference on Robot Learning (CoRL 2022), Auckland, New Zealand.

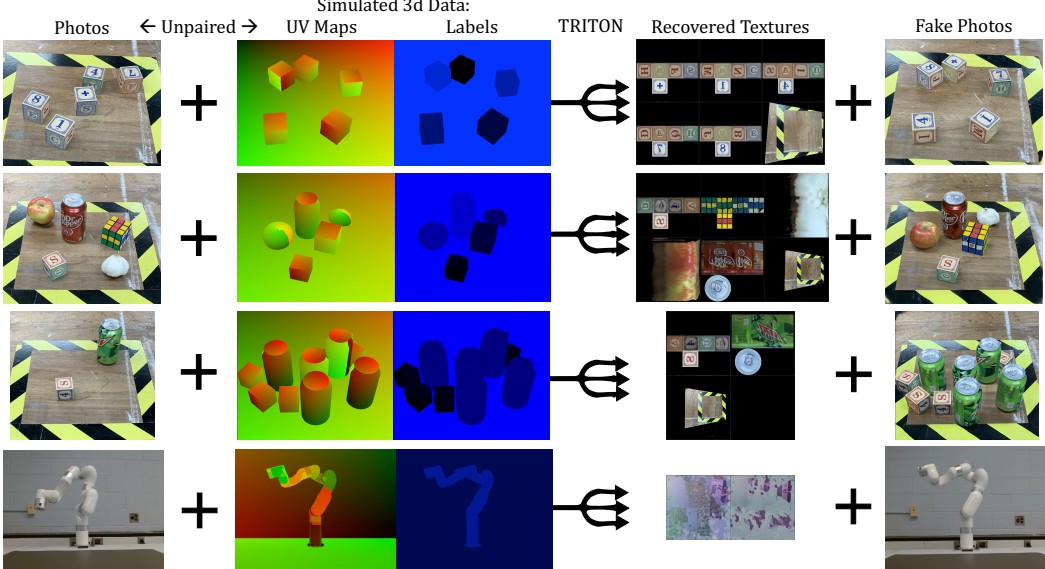

Figure 1: This is an overview of what TRITON accomplishes. TRITON learns the textures of 3D objects to help translate simulated images into photographs. It does this with unpaired data. Each row is a different dataset.

We conduct multiple experiments to confirm TRITON's advantage over prior image translation approaches commonly used for sim2real including CycleGAN [7] and CUT [8]. Importantly, we show the advantages of the proposed approach in real-robot sim2real experiments, learning the textures and training a robot policy solely based on the images generated by TRITON.

## 2 Related Works

**Unsupervised Image to Image and Video to Video Translation** Generative models have been very successful in creating images unconditionally [9, 10], creating images conditioned on text [11, 12] and creating images when trained on paired data [13]. Unpaired image translation algorithms include [14, 7, 15, 16]. Of particular interest are sim2real image to image translation algorithms such as [1, 2, 17, 18]. These algorithms aim to translate simulated images into realistic ones, for various purposes such as data augmentation or video game enhancement. In particular, Ho et al. [1], Rao et al. [2] are used to help train robots. Our paper uses an architecture based on Pfeiffer et al. [17], which is based on MUNIT [15]. Extending to video to video translation, optical flow is often used for stateful translation to improve temporal coherence [19, 20]. But they do not provide temporal coherence over long time periods.

**View Synthesis** NeRF [6] has spawned a large body of research on end-to-end view synthesis like [21, 22, 23]. Related to our work, which also uses geometry based backbone are [24, 25, 26]. However, all of these works only work with static scenes. There are NeRF-based approaches that work on dynamic scenes [27, 28, 29] but none of these were interactive, prohibiting their usage for sim2real. Menapace et al. [30] is interactive, but its conditioned on learned actions and not compatible with physics simulators that would be used to train robots. To learn the dynamics in the context of robot learning, self-supervised learning over time is an available approach [31, 32, 33, 34]. Differently, we only use the geometry without temporally aligned samples in those works.

**Neural Textures** Thies et al. [3] introduced the concept of neural textures and image translators in the context of a deferred rendering pipeline, used in other works such as [4] which is closely related to our project. Rivoir et al. [4] differs from TRITON in a few very important aspects though, the biggest being that it only synthesizes new camera views, where the only difference between each scene is the placement of the camera. In contrast, our algorithm takes in scenes where several objects are placed in different places or deformed, where one static global 3D model of the environment is not sufficient to accomplish our tasks. In addition, TRITON introduces a new type of neural texture, "neural neural texture", which is represented implicitly using a Fourier feature network [5].

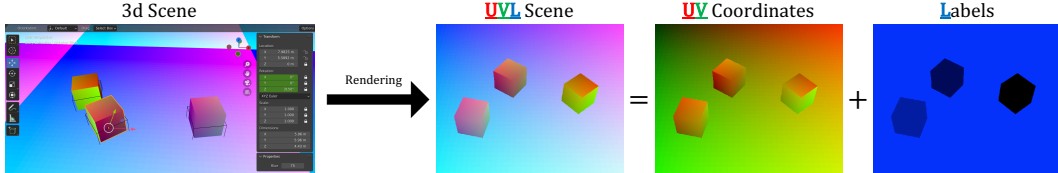

Figure 2: A scene is obtained by rendering a 3D state into an image, where the first two color channels represent $u, v$ coordinates and the third channel $l$ represents object labels.

## 3 Formulation

TRITON's goal is to turn 3D simulated images into realistic fake photographs (by training it without any matching pairs), while maintaining high surface consistency. It does this by simultaneously learning both an image translator and a set of realistic textures. These translations can be useful for downstream tasks, especially robotic policy learning from camera data, enabling sim2real.

In contrast to previous works involving neural textures [3, 4], TRITON can handle more than just camera movements: the positions of objects in the training data can be moved around or deformed as well between data samples. With high surface consistency, surfaces of translated objects will look the same even when moved around or viewed from different camera angles.

There are two components in every dataset: a set of simulated 3D scenes and a set of photographs. Datasets usually have many more scenes than photographs, since scenes can be generated automatically. A photograph is an image tensor, having $(r, g, b)$ values between 0 and 1. We denote it as $p \in [0, 1]^{H \times W \times 3}$ where 3 refers to the three $(r, g, b)$ channels, and $H, W$ are the height and width.

In this work, a scene refers to a rendered 3D simulation state - which includes the position, rotation, and deformation of every object (including the camera). We represent each 3D state with a simple $(r, g, b)$ image which we call a UVL scene, which is simple enough format that any halfway decent simulator should able to provide. A scene has the same dimensions as a photograph and is denoted as $s \in [0, 1]^{H \times W \times 3}$. However, unlike $p$, $s$'s three $(r, g, b)$ channels encode a special semantic meaning: $(u, v, l)$ as depicted in Figure 2. The channels $u$ and $v$ refer to a texture coordinate, whereas $l$ refers to a label value that identifies every object in a scene. In a given dataset, each object gets a different label. Likewise, each object is assigned a different texture. We assume that every dataset has $L$ different label values for $L$ different objects, and that we must learn $L$ different neural textures.

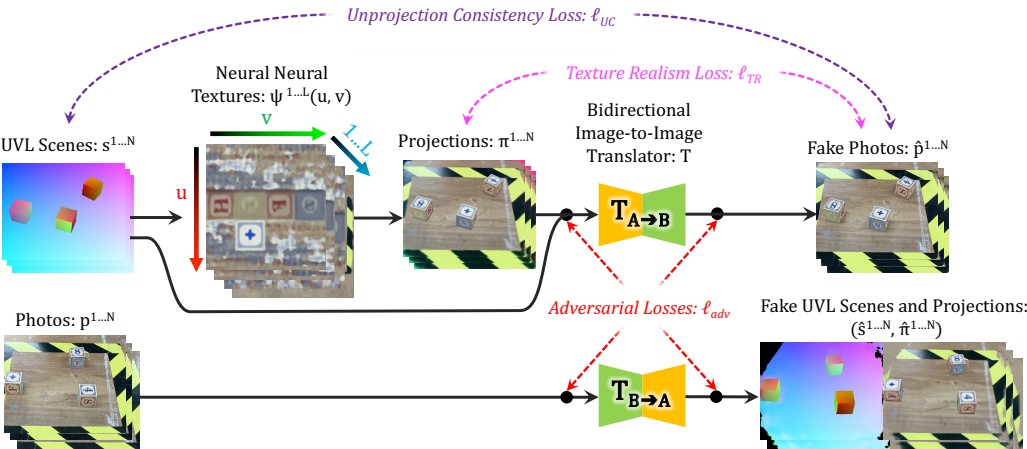

Figure 3: An overview of TRITON. Inputs are on the left, and outputs are on the right. The dashed arrows indicate losses. Although $\pi^{1...N}$ and $\hat{p}^{1...N}$ look similar, they are not identical - $\ell_{TR}$ encourages them to look as similar as possible.

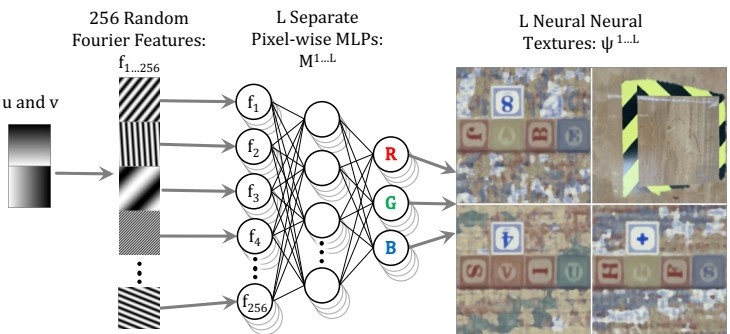

Figure 4: Details of calculating neural neural textures in Figure 3. They are fully differentiable, and represented continuously along the texture's u and v axes using Fourier feature networks.

## 4 Method

In this section, we describe how TRITON works to translate images from domain A (simulation) to domain B (photos) while maintaining surface consistency (Figure 3). TRITON introduces a learnable neural neural texture with two novel surface consistency losses to an existing image translator. In the end, TRITON is able to effectively generate photorealistic images.

### 4.1 Neural Neural Texture Projection

Instead of feeding a UVL scene $s$ directly into the image translator $T_{A \to B}$, we first apply learnable textures $\psi$ to the object surfaces, which is learned jointly with $T_{A \to B}$.

These neural neural textures are represented implicitly by a neural network that maps UV values to RGB values. For each texture $\psi^i \in \psi^{1 \dots L}$,

$$\psi^i : (u, v) \in \mathbb{R}^2 \to (r, g, b) \in \mathbb{R}^3 \tag{1}$$

Given a UVL scene $s$, we obtain projection $\pi$ by applying a texture $\psi^i \in \psi^{1 \dots L}$ to every pixel $(u, v, l) \in s$ individually, where texture index $i = I(l)$ is decided by the label value $l$ of that scene pixel. The projection $\pi$, which is an intermediate image is computed as:

$$\pi_{[x,y]} = \psi^i(s_{[x,y,1]}, s_{[x,y,2]}) \tag{2}$$

where $x \in \{1 \dots W\}$, $y \in \{1 \dots H\}$. Texture index $i = I(l)$ where $l = s_{[x,y,3]}$ and the function $I(\cdot)$ scales and discretizes $l$ into integers. Subscripts mean multidimensional indexing.

We now discuss the implementation of our neural neural networks $\psi^{1 \dots L}$. Each neural neural texture $\psi^i$ is a function consisting of two components: a multi-layer perceptron $M^i$ and a static set of 256 random spatial frequencies $f_{1 \dots 256} \in \mathbb{R}^{256 \times 2}$ (Figure 4). Given a two-dimensional $(u, v)$ vector, the spatial frequencies are used to generate a high dimensional embedding vector of $(u, v)$: $(\sin(g_1), \cos(g_1), \dots, \sin(g_{256}), \cos(g_{256}))$ . These embeddings are fed into $M^i$, where $M^i$ is a function $\mathbb{R}^{512} \to \mathbb{R}^3$ mapping that embedding to an $(r, g, b)$ color value:

$$\psi^i(u, v) = M^i(\sin(g_1), \cos(g_1), \sin(g_2), \cos(g_2), \dots \sin(g_{256}), \cos(g_{256})) \tag{3}$$

where $g_k = f_k \cdot (u, v)$. $f_k = (\alpha_k, \beta_k)$ is a two dimensional vector, where $\alpha_k, \beta_k \in \mathcal{N}(\mu = 0, \sigma = 10)$ and $\cdot$ is the dot product operator. The hyperparamters 256 and 10 are selected empirically.

In practice, we found that TRITON learns faster and more stably with our neural neural textures than it does with raster neural textures. The generated textures are also smoother and less noisy. See the supplementary material for more details.

### 4.2 Image Translation

Our image translation module $T$ is bidirectional: $T_{A \to B}$ translates sim to real (aka domain A to domain B), and $T_{B \to A}$ translates real to sim (aka domain B to domain A).

$$\hat{p} = T_{A \to B}(\pi, s) \quad \text{and} \quad (\hat{\pi}, \hat{s}) = T_{B \to A}(p) \tag{4}$$

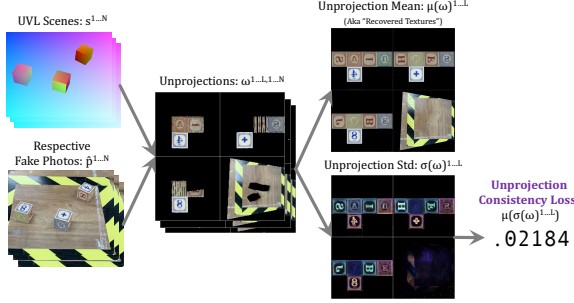

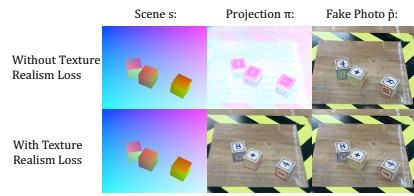

Figure 5: This is the unprojection consistency loss used in Figure 3. The Unprojection Mean is not used in any losses, but help to illustrate the content in Section 4.4.

Figure 6: The neural neural texture looks more realistic with texture realism loss enabled. Note that the blocks show different letters because of different initializations and the symmetry of a cube - both solutions are valid texture assignments.

$T$ is based on the image translation module used in Rivoir et al. [4], which is a based on a variant [17] of MUNIT [15]. $T$ uses the same network architectures as MUNIT, and also inherits its adversarial losses $\ell_{adv}$, cycle consistency losses $\ell_{cyc}$ and reconstruction losses $\ell_{rec}$. The main differences between our image translation model and MUNIT are that we use a different image similarity loss $\Omega$. Like Pfeiffer et al. [17] and Rivoir et al. [4] our style code is fixed (making it unimodal) and noise is injected into the latent codes during training to prevent overfitting. During inference however, this intermediate noise is removed and our image translation module is deterministic.

MUNIT translates images by encoding both domains $A$ and $B$ into a common latent space, then decoding them into their respective domains $B$ and $A$. In our translation module (Figure 3), we define fake photos $\hat{p} = T_{A \to B}(\pi, s) = G_B(E_A(\pi, s))$ and fake projection/UVL scenes $(\hat{\pi}, \hat{s}) = T_{B \to A}(p) = G_A(E_B(p))$, where $E_A, E_B, G_A, G_B$ are encoders and generators for domains A and B respectively.

We define an image similarity loss $\Omega$ between two images $x, y$ that returns a score between -1 and 1, where 0 means perfect similarity:

$$\Omega(x, y) = L_2(x, y) - \text{msssim}(x, y) \qquad (5)$$

This function combines mean pixel-wise $L_2$ distance (used in MUNIT) with multi-scale structural image similarity msssim, introduced in [35]. Note that this loss can also be applied to the latent representations obtained by $E_A$ and $E_B$, because like images those tensors are also three dimensional.

We have cycle consistency loss $\ell_{cyc} = \Omega\left((\pi, s), T_{B \to A}(\hat{p})\right) + \Omega\left(p, T_{A \to B}(\hat{\pi}, \hat{s})\right)$, similarity loss $\ell_{rec} = \Omega\left((\pi, s), G_A(E_A(\pi, s))\right) + \Omega\left(p, G_B(E_B(p))\right)$ (which is effectively an autoencoder loss), and content similarity loss $\ell_{con} = \Omega\left(E_A(\pi, s), E_B(\hat{p})\right) + \Omega\left(E_A(p), E_A(\hat{\pi}, \hat{s})\right)$. We also have adversarial losses $\ell_{adv}$ that come from two discriminators $D_A$ and $D_B$, targeting domains A and B respectively, using the LS-GAN loss introduced by Mao et al. [36]. In total, our image translator loss is $\ell_T = \ell_{cyc} + \ell_{rec} + \ell_{con} + \ell_{adv}$.

### 4.3 Unprojection Consistency Loss

To keep the object surfaces consistent, we impose a pixel-wise "unprojection consistency loss" by unprojecting surfaces in fake photos back into the common texture space. Given a UVL scene $s$ and its respective fake photo $\hat{p}$, the unprojection $\omega$ is obtained from assigning $(r, g, b)$ values at each pixel location $(x, y)$ of $\hat{p}$ to its corresponding $(u, v, l)$ coords given by $s$. For simplicity, we note

$$\omega^l_{[u,v]} = \mathbb{E}\,\hat{p}_{[x,y]} \quad s.t.\ (u, v, l) = s_{[x,y]}, \qquad (6)$$

where the expectation $\mathbb{E}$ means we aggregate multiple $(r, g, b)$ vectors being assigned to the same $(u, v, l)$ coordinate by averaging them. In practice, $(u, v, l)$ are real numbers between $[0, 1]$, where as each $\omega$ have to be rasterized, i.e., represented by $L$ images with a size of $(D \times D \times 3)$, where $L$ is the number of labels and $D \times D$ is the resolution of the unprojection. Therefore, we discretize and scale corresponding $(u, v, l)$ to integers so that each $(r, g, b)$ vector will be assigned to a pixel on $\omega$. For the exact implementation, please see our supplementary material.

We obtain $N$ unprojections $\{\omega^{l,i}_{[u,v]}\}^N_{i=1}$ from a batch of $N$ UVL scenes and fake photos. The unprojection consistency loss is defined as the per pixel-channel standard deviation of $\omega$ over the batch

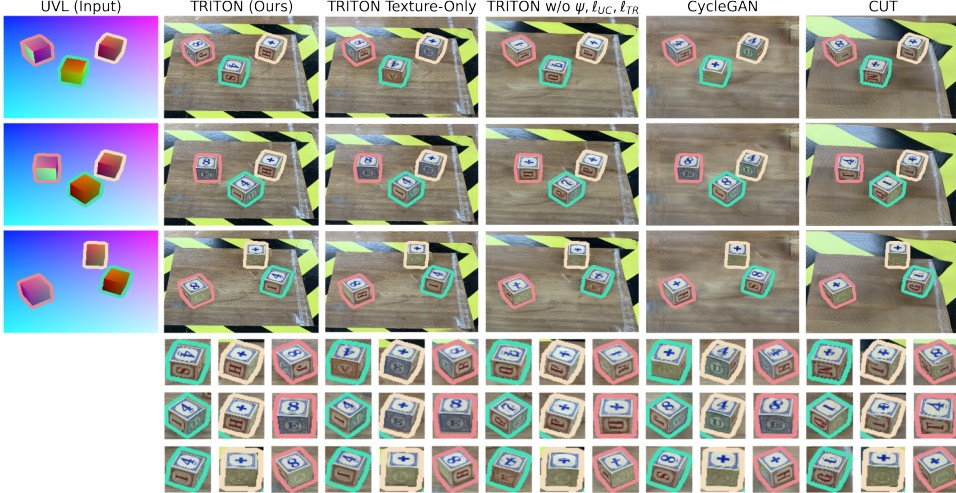

Figure 7: Image translation comparisons showing how TRITON has better temporal consistency. The first column is the UVL map as the input. The other columns are results using different image translation methods. Each row of images shows a different random placement of objects in the scene. TRITON consistently outputs high quality results over different object arrangements. "TRITON Texture-Only" stands for TRITON without two surface consistency losses $\ell_{UC}$ and $\ell_{TR}$.

$$\ell_{UC} = \frac{1}{L \times D \times D \times 3} \sum_{l,u,v,c} \sigma(\{\omega_{[u,v,c]}^{l,i}\}_{i=1}^{N}), \tag{7}$$

where $\sigma(\cdot)$ stands for the standard deviation function and $c \in \{1 \dots 3\}$ is the channel index of $\omega$. We minimize $\ell_{UC}$ to encourage unprojections to be consistent across the batch. Intuitively, if $\ell_{UC}$ were 0, it would mean the object surfaces in translations $\hat{p}$ appear exactly the same in every scene. In addition, we call the mean of unprojections $\mu(\{\omega^{l,i}\}_{i=1}^{N})$ as "Recovered Textures", visualzed in Figure 1 and Section 5.

### 4.4 Texture Realism Loss

To encourage the neural textures to look as realistic as possible, we try to make the projections look like the final output by introducing a "texture realism" loss $\ell_{TR}$. $\ell_{TR}$ is an image similarity loss that makes $\pi$ look like its translation $\hat{p}$.

$$\ell_{TR} = \Omega(\pi, \hat{p}) \tag{8}$$

$\Omega$ is the image similarity from Equation 5. Without $\ell_{TR}$, $\psi$ can look wildly different each time you train it. As seen in the top row of Figure 6, the neural texture looks very unrealistic — the colors are completely arbitrary. By adding texture realism loss $\ell_{TR}$, we make the textures $\psi$ more realistic. In practice, this makes TRITON less likely to mismatch the identity of translated objects.

## 5 Experiments

In this section, we perform two quantitative experiments: the first measures TRITON's accuracy directly, and the second measures it's sim2real capability in a real-world setting. *Please see the appendix for more interesting experiments!*

### 5.1 Datasets

We constructed two datasets AlphabetCube-3 and RobotPose-xArm to benchmark different image translators in many perspectives. In our setting, each dataset is composed of two sets of unpaired images: UVL scenes from a simulator and real photographs. Note that the images in all datasets are unpaired and UVL scenes only rely on rough 3D models of the objects in the scene without need of precisely aligning objects in the simulator to the real world. We use AlphabetCube-3 for image translation quality evaluation, and RobotPose-xArm for sim2real policy learning evaluation.

## 5.2 Image Translation Evaluation on AlphabetCube-3

AlphabetCube-3 dataset features a table with three different alphabet blocks on it. The dataset contains 300 real photos and 2000 UVL scenes from a simulator. Each photo features these three cubes with a random position and rotation. In this section, we evaluate the image translation accuracy of TRITON and other image translation methods using AlphabetCube-3. In this dataset, $L = 4$ because there are four different textures: three textures for the three cubes and one for the table.

Figure 7 shows qualitative results on the dataset. From Figure 7 we find that TRITON consistently outputs high quality results over different object arrangements. The textures keep aligned even when the position of blocks change dramatically over multiple scenes (See examples in green rectangles in Figure 7). In contrast, though MUNIT [15] and CycleGAN [7] manage to generate realistic images, the surfaces of the cubes are either consistent over scenes (See examples in the red rectangle) or replicated by mistake (See examples in the orange rectangle). Also note that the floor background of the outputs from CycleGAN are quite different from the ground truth. CUT [8] fails to generate meaningful images regarding both the foreground blocks and the background.

In quantitative experiments shown in Table 1, we manually align the simulator with 14 photos of different real world scenes and generate the UVL maps for translation. We measured the LPIPS [37] and $l2$-norm between the translated images and the real images using multiple configurations. 'Masked' means we mask out the background (which is the floor in AlphabetCube-3 dataset) and only measure the translation quality w.r.t three foreground blocks. For the 'unmasked' tests, we compare the whole image without any masking instead, and the background contributes much more to the losses due to its larger area. Similar to the qualitative results, TRITON consistently outperforms other methods under different metrics and configurations. Further ablations TRITON w/o $\psi, \ell_{UC}, \ell_{TR}$ and TRITON w/o $\ell_{UC}, \ell_{TR}$ show the importance of the new losses we introduce.

Table 1: Quantitative results on AlphabetCube-3. LPIPS [37] and $l2$-norm between the translated images and the real images are reported. We exclude backgrounds in the 'Masked' configuration. TRITON consistently outperforms other methods in different metrics and configurations

|  | Masked | | Unmasked | |
| --- | --- | --- | --- | --- |
|  | LPIPS ($\times 10^{-1}$) $\downarrow$ | L2 ($\times 10^{-2}$) $\downarrow$ | LPIPS ($\times 10^{-1}$) $\downarrow$ | L2 ($\times 10^{-2}$) $\downarrow$ |
| CycleGAN | 0.450 | 0.559 | 6.02 | 4.25 |
| CUT | 0.469 | 0.553 | 5.40 | 6.37 |
| TRITON w/o $\psi, \ell_{UC}, \ell_{TR}$ | 0.437 | 0.500 | 4.34 | 4.25 |
| TRITON w/o $\ell_{UC}, \ell_{TR}$ | 0.442 | 0.586 | 2.85 | 3.64 |
| **TRITON** | **0.286** | **0.479** | **1.17** | **1.23** |

## 5.3 Sim2real Transfer for Robot Learning

In this section we demonstrate how TRITON improves sim2real transfer for robot policy learning. To this end, we first train TRITON on a dataset consist of unpaired UVL scenes from a robot simulator (Gazebo) and photos from the real robot. Once TRITON is trained, we learn the robot policy while only utilizing the simulator. The input of the policy is the fake photo $\hat{p}$ translated from the sim-

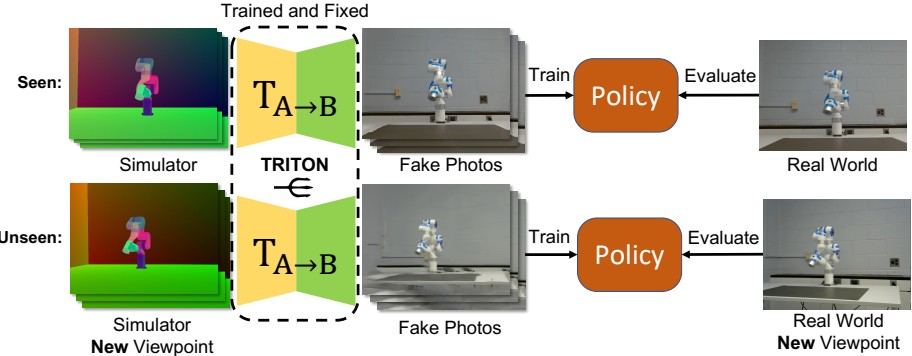

Figure 8: Sim2real framework of using TRITON. We evaluate in two settings: Seen and Unseen. In Unseen, we use a new camera viewpoint to train and evaluate the policy. This new viewpoint is not used for training TRITON.

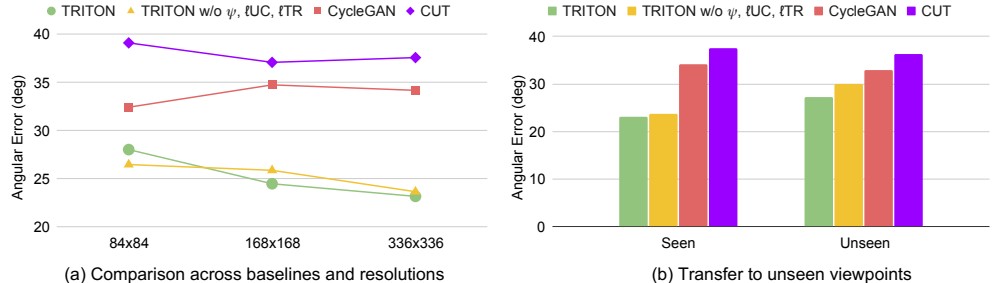

Figure 9: Evaluation results of sim2real robot learning. We compare TRITON against two baselines and one TRITON variant w/o $\psi, \ell_{UC}, \ell_{TR}$ across 3 different input image resolutions. TRITON outperforms CycleGAN [7] and CUT [8] consistently across (a) all resolutions and (b) unseen viewpoints. Results in (a) are from Seen setting only. Results in (b) are from 336x336 resolution.

ulated UVL scene using TRITON's translator $T_{A \rightarrow B}$. Finally, the trained policy is directly evaluated on the real robot, taking real photos $p$ as input.

Ideally, a good image translation model makes $\hat{p}$ similar to $p$, so that the policy trained on synthesized photos will transfer to real domain seamlessly and will have better performance. All our policies are learned with zero real-world robot interaction with the environment.

**Task** We use a robot pose imitation task for our experiments. Given the input of a photo of real robot pose, the policy outputs robot controls to replicate that target pose. We measure the angular error between the replicated and target poses as our evaluation metric: $\sqrt{\sum_j (\hat{a}_j - a_j)^2}$, where $\hat{a}_j$ and $a_j$ are replicated and target joint angles respectively, and $j$ is the joint index. We use an xArm robot which has seven joints. We also attached patterns and tapes on the robot to make its texture more challenging to model. **Robot policy and Baselines** Since the sim2real formulation allows benefiting from abundant training data using the simulator, we use behavioral cloning to train a good robot policy. The policy network is a CNN similar to [38]. We compare TRITON against two baseline image translation methods: CycleGAN [7] and CUT [8], by replacing TRITON with each baseline method respectively in the above pipeline. The robot policy learning method is same for all the methods.

**Evaluation Settings** We introduce two main evaluation settings, **Seen** and **Unseen**. In the **Seen** setting, the policy is trained and tested using the same camera viewpoint that is used during TRITON (or baseline) training. Whereas in **Unseen** setting, we introduce a camera at a new viewpoint for training and testing the policy. That is, the image translator has to generate the fake photos out of its training distribution (seen viewpoints), which becomes a challenging task for the translator. We also vary input image resolutions to show the power of photo-realistic images in higher resolutions.

**Results** Figure 9 shows the evaluation results of the sim2real transfer. Both Seen and Unseen evaluations show that TRITON outperforms baselines consistently. In Figure 9(a), we find that TRITON continuously improves from low to high resolutions due to its ability to generate more photo-realistic images compared to the baselines. From Figure 9(b), we confirm that TRITON outperforms the others also in the Unseen setting, showing that TRITON learns image translations that better generalize to new viewpoints. TRITON outperforms TRITON w/o $\psi, \ell_{UC}, \ell_{TR}$ consistently except at the lowest resolution. This again shows the effectiveness of $\psi, \ell_{UC}$ and $\ell_{TR}$ to generate photo-realistic images, and such high quality images are important in higher resolutions for sim2real transfer.

In Appendix in the supplementary material, we provide more experimental results.

## 6    Limitations

TRITON relies on 3D models of objects in order to generate photo-realistic images, making it less applicable when the geometry of an environment is unknown. Acquiring such 3D models could be challenging in the wild, especially in robots allowed to roam the world. TRITON is not able to handle transparent objects, because the UVL format is opaque. Generated shadows, though visually realistic, are not physically accurate. Moreover, the probability of incorrectly matching textures to objects increases as the number of object classes increases.

**Acknowledgments**

We thank Srijan Das and Kanchana Ranasinghe for valuable discussions. This work is supported by Institute of Information & communications Technology Planning & Evaluation (IITP) grant funded by the Ministry of Science and ICT (No.2018-0-00205, Development of Core Technology of Robot Task-Intelligence for Improvement of Labor Condition. This work is also supported by the National Science Foundation (IIS-2104404 and CNS-2104416).

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
