# OpenReview forum: "TRITON: Neural Neural Textures for Better Sim2Real"
_robot-learning.org/CoRL/2022/Conference — CoRL 2022 Poster_

### Official Review · Reviewer_uVwa · 2022-07-08

**Originality:** Good
**Technical Quality:** Good
**Clarity Of Presentation:** Good
**Impact:** 4

**Recommendation:**

Weak Accept: I recommend accepting the paper, but will not argue for my recommendation if the majority of other reviewers have a different opinion.

**Summary:**

The authors propose a new method called TRITON in the domain of image translation tasks. Given a real-world photograph, the model can learn the texture of each object and translate a simulated scene into a photorealism image. The textures of objects are generated using differentiable rendering parametrized by neural networks (named neural neural texture). Specifically, the input (u,v map coordinates) of neural rendering goes though a fourier feature network in a high dimensional space, which is parametrized afterwards with $L$ MLPs and the outputs are pixel-wise rgb value. Moreover, in order to maintain the temporal consistency of objects between frames, the authors propose a surface consistency loss. Another texture realism loss and adversarial loss are included to close the gap between simulator and real-world. The experiments show that TRITON can surpass prior work CycleGAN and CUT by a large margin.

**Issues:**

1. The experimental results on various shapes of objects are missing, e.g., like the example showed in fig.1 row 2&3.
2. The details of experiments on RobotPose-xArm.

**Quality Of The Limitations Section:**

Limitations are addressed clearly

**Reviewer Expertise:**

4: The reviewer is confident but not absolutely certain that the evaluation is correct

**Robotics Focus:**

Sufficient demonstration on hardware

**Strengths And Weaknesses:**

**Strengths**:
1. The authors combine fourier network and neural rendering, which gives continuous and smooth representations over the UV space for the texture rendering. The authors also claim that it is faster and more stable compared to raster neural textures.
2. To keep the consistency of object surfaces, the authors propose a novel loss, which unprojects surfaces of a batch of fake images to a common texture space. Simply reducing the standard deviation of the pixel-wise $w$ imposes the surface consistency.
3. The authors create two datasets to evaluate their method, namely AlphabetCube-3 and RobotPose-xArm. In particular, the experiments on RobotPose-xArm demonstrate the image translator on unseen camera viewpoint. The generated video looks promising and could potentially help sim2real policy learning and alleviate human efforts in labelling.

**Comments**:
1. Does $L$ also include the background, e.g. $L$ in AlphabetCube-3 is 3 or 4?
2. Can you elaborate on more details of the RobotPose-xArm dataset, e.g. how many photographs, simulated scenes, $L$ and training details?
2. I think TRITON is only working on known objects in the training dataset. Is that possible to render textures of unseen objects (e.g. duck which is not shown during training) given an existing label (e.g. the label of "apple") using TRITON? Or it might fail because of different geometric features.
3. In Fig. 1 row 2, we see multiple different objects. I think you already tested on them, why not include their results in the experimental section but only AlphabetCube-3? This is a critical missing point in my opinion.
4. Do you think $L$ could be variable? Currently, the number of MLPs is implied by the number of objects. Therefore, TRITON is not scalable given large numbers of different objects.
5. I think the UVL label along with the value of $L$ are still a strong prior knowledge since they reveal the information between photographs and scenes, which is limited because in practice, predefined object labels sometimes are not accessible. It would be interesting to see an extended work: new images can be generated given UVL scenes by learning the textures from photographs, where the objects in scenes and photographs are different or even the objects in scenes are unknown during training.

**minor**:
A margin space is missing between the caption of fig.4 and the following paragraph.



**Summary Of Recommendation:**

Overall I think this work is well-motivated and technically sound. The results look promising which support potential applications for robot learning, e.g., imitation learning or policy learning. Some minor changes could be made to improve the clarity, e.g., more details about the experiments on RobotPose-xArm, an experimental results on various objects like in fig.1 row 2 instead of only cubes.

---

> ### Author Response · Authors · 2022-08-27
> **Response to Reviewer uVwa (Part 2/2)**
>
> > Do you think L could be variable? Currently, the number of MLPs is implied by the number of objects. Therefore, TRITON is not scalable given large numbers of different objects.
>
> Yes and no, depending on what it means for L to be variable. If we train on a dataset with L=5 different objects, and then we remove some of them during evaluation, there should be no problem. It’s true that TRITON at this point isn't scalable if you plan to train on a dataset containing a very large L value. However, if you wish to train for a large number of objects, you can train TRITON on multiple datasets with smaller values of L, then combine their learned textures together into one large model. This works because the neural textures are regularized to look realistic, making them non-arbitrary and modular.
>
>
> > I think the UVL label along with the value of  are still a strong prior knowledge since they reveal the information between photographs and scenes, which is limited because in practice, predefined object labels sometimes are not accessible.
>
> We would like to clarify that in a sim2real setting it is very convenient to get the UVL label from the simulator and our method does not require any labels from the real photographs.
> > It would be interesting to see an extended work: new images can be generated given UVL scenes by learning the textures from photographs, where the objects in scenes and photographs are different or even the objects in scenes are unknown during training.
> >
> We thank you for the suggestion, and we will explore this further in the future work.
>
>
> > minor: A margin space is missing between the caption of fig.4 and the following paragraph.
>
> Thank you for pointing out the margin spacing, we'll fix that!
>
> Please take a look at our robotic experiment videos:
> - https://youtu.be/IwAs420hY6A
> - https://youtu.be/JXNLwWoAlgQ
>
> We explain them in more detail in the attached `rebuttal.pdf`

---

> ### Author Response · Authors · 2022-08-27
> **Response to Reviewer uVwa (Part 1/2)**
>
> We thank the reviewer for the comment, and we would like to address all the concerns.
>
> > Does L also include the background?
>
> Yes, it does! For instance, in AlphabetCube-3, L=4 - there are 4 textures that must be learned, including the background table texture.
>
>
> > Can you elaborate on more details of the RobotPose-xArm dataset, e.g. how many photographs, simulated scenes, L and training details?
>
> We collected 891 different robot arm poses in the RobotPose-xArm dataset. Each pose was collected from 3 different camera views. The dataset was collected in pairs of simulated and real images -- for evaluation purposes -- resulting in 891*3 simulated images and 891\*3 real images in total. In this dataset, L=2, which are the textures of the background and the robot arm.
>
> The architecture of the policy network can be denoted as C(8, 32, 4)-C(4, 64, 2)-C(3, 128, 2)-C(3, 256, 2)-C(3, 256, 1)-FC(action size)-Tanh(), where C(k, c, s) stands for a convolutional layer, k is the kernel size, c is the number of channels and s is the stride. FC(s) is one fully connected layer with an output size of s. Each convolutional layer is followed by a ReLU activation.
> The network is trained by Adam using learning rate=1e-4 with batch size=4 for 10 epochs.
>
>
> > I think TRITON is only working on known objects in the training dataset. Is that possible to render textures of unseen objects (e.g. duck which is not shown during training) given an existing label (e.g. the label of "apple") using TRITON? Or it might fail because of different geometric features.
>
> Yes, TRITON only works on objects it was trained on.
> It's possible in the sense that TRITON won't crash, but won't look very good. In our new dataset, we decided to include both an apple and a rubber duck! See this image: [https://i.imgur.com/PVJhvSd.png](https://i.imgur.com/PVJhvSd.png). To further answer this question, we’ve also made an analysis in our submitted `rebuttal.pdf` that demonstrates what happens when you shuffle the labels, but then let it train for while afterwards. The result looks quite interesting. (Please check section C.1)
>
>
> > In Fig. 1 row 2, we see multiple different objects. I think you already tested on them, why not include their results in the experimental section but only AlphabetCube-3? This is a critical missing point in my opinion.
>
> We conducted additional experiments with multiple objects, also testing it on a real robot. Figures/Tables in the attached new `rebuttal.pdf` shows the results:
>
> 1. Section B.2: A real-world robotic arm experiment, where the arm learns to position itself over five different randomly positioned objects on a table. We show that TRITON enables the behavioral cloning algorithm to learn policies that transfer better to the real world. See subsection titled “Robot Manipulator Experiment - Reacher” in the new `rebuttal.pdf` for more details.
>
> 2. Section B.3: A synthetic american flag that flutters in the wind, demonstrating how TRITON performs on deformable objects. We show that, by a large margin, TRITON outperforms other baselines in accuracy. Because the ground truth is synthetic, we can accurately measure the output’s pixel accuracy on every frame. See subsection titled “American Flag Experiment” in the new `rebuttal.pdf` for more details.
>
> 3. Section B.4 A synthetic set of three rigid 3d objects including a treasure chest, an avocado and steve from minecraft. Again, we show that TRITON produces results that are closer to the ground truth than other baselines. See subsection titled “Synthetic Object Experiment” in the new `rebuttal.pdf` for more details.
>
> These new experiments have multiple baselines including CUT, CycleGAN, and MUNIT.
>
> We also provide qualitative results for the datasets mentioned in figure 1 row 2 in the form of an animation. See [https://youtu.be/9zC4-ggmN7Q](https://youtu.be/9zC4-ggmN7Q) and [https://youtu.be/-GSixT4shxY](https://youtu.be/-GSixT4shxY) from the original appendix (figure 15). As you watch them move, notice how TRITON is more temporally consistent than the baselines.
> The reason we focused on AlphabetCube-3 instead of those other datasets shown in figure 1 is because it's easier to get a ground truth for three cubes than it is for the other datasets such as the ones with soda cans – it’s much harder to match five objects to a ground truth photo than it is to match three cubes.

---

### Official Review · Reviewer_9MxW · 2022-07-31

**Originality:** Good
**Technical Quality:** Very Good
**Clarity Of Presentation:** Very Good
**Impact:** 3

**Recommendation:**

Weak Accept: I recommend accepting the paper, but will not argue for my recommendation if the majority of other reviewers have a different opinion.

**Summary:**

The authors propose a new method TRITON that can transform textures from real images to synthetic scenes in an unpaired fashion. The approach requires UVL presentations for sim2real texture transfer. The authors conducted experiments to show the strength of the approach in robotic experiments and image translation evaluation.

**Issues:**

Please address my concerns above.

Is it a typo in the title? "Neural Neural"?

**Quality Of The Limitations Section:**

Additional details required

**Reviewer Expertise:**

4: The reviewer is confident but not absolutely certain that the evaluation is correct

**Robotics Focus:**

Sufficient demonstration on hardware

**Strengths And Weaknesses:**

Strength
+ The presented approach is technically interesting. The technical solution is clearly well-written.
+ There are clear improvements with the presented approach over other methods such as CycleGAN and CUT.
+ There are hardware experiments to demonstrate the strength of the proposed method.

Weaknesses
- There is no analysis or experiment about temporal consistency. Although the authors claim that the approach can ensure consistency between frames statelessly.
- I think the presented approach can *not* handle cases with shadows. The inherent representation limits such a possibility. If this is the case, this should be mentioned in the limitation section.
- I think MUNIT is also a baseline for comparisons. Can authors also evaluate this method?
- The approach requires UVL representations for the 3D scenes, which make it less generally applicable.

**Summary Of Recommendation:**

I think the approach is interesting, but more experiments and discussions are needed.

Thanks for the rebuttal. It basically addresses my concerns. I will keep my score.

---

> ### Author Response · Authors · 2022-08-27
> **Response to Reviewer 9MxW**
>
> We thank the reviewer for the comment, and we would like to address all the concerns.
>
> > There is no analysis or experiment about temporal consistency. Although the authors claim that the approach can ensure consistency between frames statelessly.
>
> We would like to clarify. Figure 7 demonstrates temporal consistency by showing how the patches between objects look the same from different views as they move around. We are able to observe that TRITON does a good job compared to our baselines (CycleGAN, MUNIT, CUT, and an ablation). We will make this more explicit in the final version of the paper. It can also be seen in the video in `TRITON_Appendix.pdf` from the supplementary material, in A.4.2 Figure 14 at [https://youtu.be/-GSixT4shxY](https://youtu.be/-GSixT4shxY) (the link to the video in the appendix was wrong - this is the correct video). Apology for the confusion - we will explicitly state that this figure demonstrates temporal consistency and fix the link in the final paper.
>
> In addition to these figures, our new experiments come with several new animations that show qualitatively how TRITON has better temporal coherence than the baselines.
>
>
> > I think the presented approach can handle cases with shadows. The inherent representation limits such a possibility. If this is the case, this should be mentioned in the limitation section.
>
> It’s true that TRITON can’t generate physically accurate shadows, as it doesn’t have direct access to the underlying 3d geometry. We will add this discussion to the limitation section. However, TRITON does generate fake shadows and lighting effects as a result of the GAN process in the image translation module. For a more detailed analysis on shadows generated by TRITON, see Section C.2 in the new `rebuttal.pdf` attached for this rebuttal.
>
>
> > I think MUNIT is also a baseline for comparisons. Can authors also evaluate this method?
>
> Thanks for the suggestion,  in our new experiments, we compared directly against MUNIT. The newly added baselines labeled MUNIT use only L2 losses on the embeddings, just as was done in the original MUNIT paper (we still use fixed style codes which turned out to perform better).
>
> The baseline “Triton without textures” we used throughout the paper in fact is very similar to MUNIT, as TRITON’s codebase is a modified version of MUNIT. There are a few small differences however: instead of allowing the style code to vary, we fix it to a single value. Rivoir et al. [17] discovered that MUNIT’s styles harmed the temporal consistency of the translation results, so for simplicity we also disable them. Additionally, we made the baselines to use both L2 and MSSSIM losses on the image embeddings, modifying it to match what TRITON uses.
>
>
> > The approach requires UVL representations for the 3D scenes, which make it less generally applicable.
>
> We would like to clarify that UVL representations are very generalizable, because any simulator that can output the positions of 3d objects in a scene is able to create a UVL scene within a 3d renderer. There are many possible rendering front-ends such as Blender, Unity, and Pytorch3d - all of which can take 3d information and turn it into a UVL scene. In fact, many simulators like Gazebo, Pandas3d, Unity, and Isaac Gym have 3d rendering capabilities out of the box and can be trivially modified to produce UVL scenes by simply applying UVL images such as [https://i.imgur.com/4oH9w5V.png](https://i.imgur.com/4oH9w5V.png) or  [https://imgur.com/yfgZefg](https://imgur.com/yfgZefg) onto the 3d objects as a texture.
>
>
> > Is it a typo in the title? "Neural Neural"?
>
> We apologize for the confusion. It's not a typo. Previous works called the learnable textures "neural textures", which were parametrized by a discrete grid of differentiable texels. In contrast, we call our learnable textures as neural neural textures, because our textures are represented implicitly as a neural network function, parameterized continuously over UV space. Using this representation instead of using discrete texels allows TRITON to learn faster and yields better results.
>
> Please take a look at our robotic experiment videos:
> - https://youtu.be/IwAs420hY6A
> - https://youtu.be/JXNLwWoAlgQ
>
> We explain them in more detail in the attached `rebuttal.pdf`

---

### Official Review · Reviewer_agLh · 2022-08-01

**Originality:** Good
**Technical Quality:** Good
**Clarity Of Presentation:** Very Good
**Impact:** 3

**Recommendation:**

Weak Reject: I recommend rejecting the paper, but will not argue for my recommendation if the majority of other reviewers have a different opinion.

**Summary:**

This paper proposed a 3D to 2D, learned rendering pipeline that can adapt the textures given a set of images. The learned rendering pipeline can then be used to generate synthetic dataset for robotic sim2real transfer.


**Issues:**

- Why do authors name the method “Neural Neural Texture”? Why are there two “Neural”?
- Line 35, *”learning the textures and training a robot policy solely based on the images generated by TRITON.”*
  Does this mean TRITON learns the textures from images generated by itself, or this is a typo?
- Line 73, *”the positions of objects in the training data can be moved around or deformed as well between data samples”*
  Do authors have results on deformable objects?


**Quality Of The Limitations Section:**

Limitations are addressed clearly

**Reviewer Expertise:**

3: The reviewer is fairly confident that the evaluation is correct

**Robotics Focus:**

Relevant but unlikely to deploy to hardware in near future

**Strengths And Weaknesses:**


### Strengths
**Clarity**

I think the figures are very illustrative and beautiful.

**Quality**
- The proposed method is technically sound.
- The limitation section is honest and complete.

### Weaknesses

**Missing Citation**

The following paper uses 3D model + domain randomization to achieve camera-to-robot pose estimation is highly relevant as it has a similar real-world demo.
- [1] Camera-to-Robot Pose Estimation from a Single Image, ICRA 2020

**Baselines**

I think it’s unfair to compare the proposed method with CycleGAN and CUT as they don’t have access to 3D scenes. It would be more suitable to evaluate against baselines that actually use the 3D information, e.g. [1] listed above.

**Significance**

- Since the focus on this paper is sim2real, I think the real-world experiments are very important. However, the real-world experiments presented in this paper have low significance because a) it’s not a real task that roboticists encounter often and b) there are methods which can solve this problem, e.g., [1]. I believe unique, novel robotic abilities enabled by TRITON in the real world are lacking.
- For image translation evaluation, I think the AlphabetCube-3 dataset may be too simple (as the shape is regular and the object itself is rigid). Additional results on non-rigid (e.g., deformable) objects will make the task more challenging.

**Clarity**
- The name of each method shown at the top of Figure 7 is blurry.
- Line 96, the image caption and main text are too close to each other.
- I am not sure why the proposed method has two Neural in its name.

**Typo**
- Line 34, Important -> Importantly

- Line 108, neural neural networks -> neural networks


**Summary Of Recommendation:**

I recommend weak rejection because a) I think comparing to baselines that can only access 2D information is not fair, b) the paper focuses on sim2real but the real-world experiments are not convincing.

---

> ### Author Response · Authors · 2022-08-27
> **Response to Reviewer agLh**
>
> We thank the reviewer for the comment, and we would like to address all the concerns.
>
> > The following paper uses 3D model + domain randomization to achieve camera-to-robot pose estimation is highly relevant as it has a similar real-world demo.
>
> Thank you for pointing this paper out, we will cite it in our paper under the section involving our robotic pose imitation experiment, and discuss it further. The difference is that the main purpose of our paper is sim2real, whereas this paper’s focus is about pose estimation. And the method in the paper you mentioned takes different types of information compared with our method.
>
> > I think it’s unfair to compare the proposed method with CycleGAN and CUT as they don’t have access to 3D scenes. It would be more suitable to evaluate against baselines that actually use the 3D information, e.g. [1] listed above.
>
> We would like to clarify that both TRITON and all our baselines (CycleGAN, MUNIT, CUT, and TRITON ablations) use the exact same input tensor. TRITON doesn't use any information that CycleGAN and CUT don't also have access to: this is a sim2real setting where all algorithms take in three-channel images from the simulator, and output three-channel images of the same size.
>
> > Why do authors name the method “Neural Neural Texture”? Why are there two “Neural”?
>
> We apologize for the confusion. Previous works called these learnable textures "neural textures", and were parametrized by a discrete grid of differentiable texels. In contrast, we call our learnable textures as neural neural textures, because our textures are represented implicitly as a neural network function, parameterized continuously over UV space. Using this representation instead of using discrete texels allows TRITON to learn faster and yields better results.
>
>
> > Line 35, ”learning the textures and training a robot policy solely based on the images generated by TRITON.” Does this mean TRITON learns the textures from images generated by itself, or this is a typo?
>
> You're right, this sentence was a bit unclear - we can reword this sentence: "Additionally, show how our approach can aid robotic sim2real experiments, by training TRITON to make realistic images from visually unrealistic 3d simulations, then training a robot policy solely based on the images generated by TRITON."
>
>
> > Line 73, ”the positions of objects in the training data can be moved around or deformed as well between data samples” Do authors have results on deformable objects?
>
> In the paper, we show TRITON with rigid objects and a robotic arm. Here, in Section B.3 in the newly attached `rebuttal.pdf`, we conducted additional experiments with very deformable objects such as a synthetic american flag that flutters in the wind. The result of this experiment demonstrated how TRITON performs on deformable objects. We show that, by a large margin, TRITON outperforms other baselines in accuracy. Because the ground truth is synthetic, we can accurately measure the output’s pixel accuracy on every frame. More details can be found in the attached `rebuttal.pdf` Section B.3.
>
> > The name of each method shown at the top of Figure 7 is blurry.
> > Line 96, the image caption and main text are too close to each other.
> > Line 34, Important -> Importantly
> > Line 108, neural neural networks -> neural networks
>
> Thank you for pointing out these issues - we'll fix them in the final version of the paper.
>
> Please take a look at our robotic experiment videos:
> - https://youtu.be/IwAs420hY6A
> - https://youtu.be/JXNLwWoAlgQ
>
> We explain them in more detail in the attached `rebuttal.pdf`

---

### Official Review · Reviewer_z1oF · 2022-08-10

**Originality:** Very Good
**Technical Quality:** Good
**Clarity Of Presentation:** Very Good
**Impact:** 4

**Recommendation:**

Weak Accept: I recommend accepting the paper, but will not argue for my recommendation if the majority of other reviewers have a different opinion.

**Summary:**

This paper presents TRITON, a novel algorithm for generating textures for images to aid in sim2real transfer. Unlike previous methods, TRITON claims not only to maintain surface consistency across camera positions, but object positions and deformations.

The method is as follows: a "neural neural" texture projection, where scenes are mapped to textures that are learned and also represented as neural networks. The textures are applied to image pixels to obtain the projections. These are given to an image-to-image translator to get synthetic but realistic photos. The image translator can also turn real photos into synthetic photos. Losses require that projected surfaces transformed back into the texture space maintain consistency, that projections look like images, and other standards.

Experiments include testing image translation and sim2real. TRITON outperforms cycleGAN alone and CUT on image translation, and improves performance on a pose imitation sim2real transfer task.

**Issues:**

- More experimentation if possible, along with analysis of what the results say about the intuition behind this approach
- Baselines if possible
- Clarity around some of the equations, especially the losses

**Quality Of The Limitations Section:**

Limitations are addressed clearly

**Reviewer Expertise:**

2: The reviewer is willing to defend the evaluation, but it is quite likely that the reviewer did not understand central parts of the paper

**Robotics Focus:**

Sufficient demonstration on hardware

**Strengths And Weaknesses:**

**Strengths**
- Paper is fairly well-presented. There are many moving parts and it's certainly not a simple method, but each part seems well-motivated and is explained clearly.
- The approach is creative, nonobvious, and novel to my knowledge
- Evaluation is posed well. It is helpful to separate out image translation specifically, even though the primary claim is sim2real, because the algorithm structure is complicated enough to be a bit mystifying if all we learn about is the output. I appreciate the investigation in the middle of the overall pipeline.

**Weaknesses**
- Explanation of GAN-related losses gets very confusing. It's less problematic because they are partially taken from prior work, but the final paragraph of 4.2 could be much clearer.
- It would be interesting to get more intuition/evidence for why representing the texture as a neural function is beneficial, i.e. why it creates a performance gain. It's not hard to discern but I think intuition into new methods is always worth talking about.
- I realize that sim2real is expensive, but more testing would be more convincing. Baselines, which might be easier to run, would potentailly be even more helpful.

**Summary Of Recommendation:**

I think this paper successfully designs and validates a sophisticated method for generating textures that can aid in sim2real transfer. While a more robust experimental/baseline suite would help, the claims w.r.t. novel object positions and deformations seem compelling. Ultimately I don't have much domain knowledge about texture generation, so I find it somewhat hard to gauge this work against its peers, but for sim2real it does seem like a useful result with good execution.

---

> ### Author Response · Authors · 2022-08-27
> **Response to Reviewer z1oF**
>
> We thank the reviewer for the comment, and we would like to address all the concerns.
>
> >  Explanation of GAN-related losses gets very confusing. It's less problematic because they are partially taken from prior work, but the final paragraph of 4.2 could be much clearer.
>
> Thank you for your feedback, we will rewrite them in the final paper to make it more clear and concise.
>
>
> > It would be interesting to get more intuition/evidence for why representing the texture as a neural function is beneficial, i.e. why it creates a performance gain. It's not hard to discern but I think intuition into new methods is always worth talking about.
>
> Please check the supplementary material, `TRITON_Appendix.pdf` part A.1 in Figure 10 as we tried to describe it here. As shown in the experiment/example, the previous neural texture (without the function representation) tends to suffer compared to TRITON. With the conventional rasterized texture representation, the probability of each texel receiving a gradient is inversely proportional to the texture’s resolution. High resolution textures don’t receive enough gradients, and thus fail to produce realistic colors. Meanwhile, making texture representation to be low resolution receives more gradients per texel, but the result doesn’t hold enough information to produce realistic translations.
>
>
> > I realize that sim2real is expensive, but more testing would be more convincing. Baselines, which might be easier to run, would potentailly be even more helpful.
>
> Following the suggestion from the reviewer, we conducted additional experiments with multiple objects, also testing it on a real robot. Figures/Tables in the attached new `rebuttal.pdf` shows the results:
>
> 1. Section B.2: A real-world robotic arm experiment, where the arm learns to position itself over five different randomly positioned objects on a table. We show that TRITON enables the behavioral cloning algorithm to learn policies that transfer better to the real world. See subsection titled “Robot Manipulator Experiment - Reacher” in the new `rebuttal.pdf` for more details.
>
> 2. Section B.3: A synthetic american flag that flutters in the wind, demonstrating how TRITON performs on deformable objects. We show that, by a large margin, TRITON outperforms other baselines in accuracy. Because the ground truth is synthetic, we can accurately measure the output’s pixel accuracy on every frame. See subsection titled “American Flag Experiment” in the new `rebuttal.pdf` for more details.
>
> 3. Section B.4 A synthetic set of three rigid 3d objects including a treasure chest, an avocado and steve from minecraft. Again, we show that TRITON produces results that are closer to the ground truth than other baselines. See subsection titled “Synthetic Object Experiment” in the new `rebuttal.pdf` for more details.
>
> These new experiments have multiple baselines including CUT, CycleGAN, and MUNIT.
>
>
> > More experimentation if possible, along with analysis of what the results say about the intuition behind this approach
>
> Please see the above response, as we add more experiments and analyses in the new attached `rebuttal.pdf`
>
> > Clarity around some of the equations, especially the losses
> >
> Thanks for the suggestion, we will clarify them in the final paper.
>
> Please take a look at our robotic experiment videos:
> - https://youtu.be/IwAs420hY6A
> - https://youtu.be/JXNLwWoAlgQ
>
> We explain them in more detail in the attached `rebuttal.pdf`

---

### Meta-Review · Area_Chair_cKiy · 2022-08-15

**Recommendation:** Accept (Poster)
**Confidence:** 3

**Metareview:**

Scores: z1oF: Weak Accept, agLh: Weak Reject, 9MxW: Weak Accept, uVwa: Weak Accept

Quality: The paper provides a novel algorithm for generating temporally consistent sim2real transfers, which is an important contribution. The authors showed  in baseline comparisons and an ablation study that the proposed methods outerperforms existing SoTA algorithms. However, the video seems to show that the learned policy has low precision for the reaching task (see Cons).

 Clarity: The paper has been improved in the revision. New experiments with rigid and soft objects and an additional baseline have been included. The term “neural neural texture” has been clarified.

 Originality: The paper is interesting, and the proposed method is novel and creative.

Significance: Generating temporally consistent sim2real transfers are significant for robot learning, and the presented method provides a technically sound approach to address this important problem.

 Pros:

- The problem is well-motivated, and the approach is novel and creative.

 Cons:

 - The video ((https://youtu.be/IwAs420hY6A) seems to show that the learned policy has low precision for the reaching task (reaching for the apple at the time 0:10, pepper at the time 0:55, and apple again at the time 2:02). Can this be improved?
- The approach requires UVL representations for the 3D scenes, which may make it less applicable.

**Best Paper Nomination:**

No

---

> ### Author Response · Authors · 2022-08-27
> **Response to Meta Review (cKiy)**
>
> Thank you for the suggestions/comments. We tried our best to address the AC’s and the reviewers’ concerns. This includes:
> 1. We have conducted a new sim2real experiment with multiple objects, learning an action policy for a robot manipulator.
> 2. We evaluate our method TRITON and other baselines on additional datasets. We would like to clarify that all the methods take the exact same inputs.
> 3. We included an additional baseline requested by the reviewers (i.e., MUNIT) in our experiments.
> 4. We clarified the term “neural neural texture”.
>
> Please take a look at our robotic experiment videos:
> - https://youtu.be/IwAs420hY6A
> - https://youtu.be/JXNLwWoAlgQ
>
> We explain them in more detail in the attached `rebuttal.pdf`